# Transcriptomic analysis of cellular senescence induced by ectopic expression of ATF6α in human breast cancer cells

Ju Won Kim[1]☯, So-Hyun Bae[2]☯, Yesol Moon 📀[1,3], Eun Kyung Kim📀[1,3], Yongjin Kim[1,3], Yun Gyu Park[1,3], Mi-Ryung Han[2]*, Jeongwon Sohn📀[1,3]*

1 Department of Biochemistry and Molecular Biology, Korea University College of Medicine, Seoul, South Korea, 2 Division of Life Sciences, College of Life Sciences and Bioengineering, Incheon National University, Incheon, South Korea, 3 Korea Institute of Molecular Medicine and Nutrition, Seoul, South Korea

☯ These authors contributed equally to this work.
* biojs@korea.ac.kr (JS); genetic0309@inu.ac.kr (MRH)

## Abstract

### Background

The transcriptomic profile of cellular senescence is strongly associated with distinct cell types, the specific stressors triggering senescence, and temporal progression through senescence stages. This implies the potential necessity of conducting separate investigations for each cell type and a stressor inducing senescence. To elucidate the molecular mechanism that drives endoplasmic reticulum (ER) stress-induced cellular senescence in MCF-7 breast cancer cells, with a particular emphasis on the ATF6α branch of the unfolded protein response. We conducted transcriptomic analysis on MCF-7 cells by ectopic expression of ATF6α.

### Methods

Transcriptomic sequencing was conducted on MCF-7 cells at 6 and 9 hours post senescence induction through ATF6α ectopic expression. Comprehensive analyses encompassing enriched functional annotation, canonical pathway analysis, gene network analysis, upstream regulator analysis and gene set enrichment analysis were performed on Differentially Expressed Genes (DEGs) at 6 and 9 hours as well as time-related DEGs. Regulators and their targets identified from the upstream regulator analysis were validated through RNA interference, and their impact on cellular senescence was assessed by senescence-associated β-galactosidase staining.

### Results

ATF6α ectopic expression resulted in the identification of 12 and 79 DEGs at 6 and 9 hours, respectively, employing criteria of a false discovery rate < 0.05 and a lower fold change (FC) cutoff |log2FC| > 1. Various analyses highlighted the involvement of the UPR and/or ER Stress Pathway. Upstream regulator analysis of 9 hour-DEGs identified six regulators and eleven target genes associated with processes related to cytostasis and 'cell viability and

**Data Availability Statement:** All relevant data are within the manuscript and its Supporting Information files. In addition, We have deposited our RNAseq data files in the Gene Expression

Omnibus (GEO) database. The data generated in this study is available in the GEO database under accession code GSE270286. The URL for accessing the data is https://www.ncbi.nlm.nih.gov/geo/query/acc.cgi?acc=GSE270286.

**Funding:** This work was supported by the Basic Science Research Program through the National Research Foundation of Korea (NRF) funded by the Ministry of Education (Grant 2018R1D1A1B07048901 to J.S.) and the Ministry of Science and Technology Information and Communication (Grant 2021R1F1A1063994 to J. S.) of the South Korean government, and the Korea University (Grant K1824361 to J.S.) and the Incheon National University Research Grant in 2021 (to M.-R. H.). The funders had no role in study design, data collection and analysis, decision to publish, or preparation of the manuscript.

**Competing interests:** The authors have declared that no competing interests exit.

**Abbreviations:** ATF6, activating transcription factor 6α; ER, Endoplasmic Reticulum; UPR, unfolded protein response; PR, progesterone receptor; HER2, human epidermal growth factor receptor 2; TIS, therapy-induced cellular senescence; DEG, differentially expressed genes; siRNA, small interfering RNA; SA-β-gal, senescence associated-β-galactosidase.

cell death of connective tissue cells.' Validation confirmed the significance of MAP2K1/2, GPAT4, and PDGF-BB among the regulators and DDIT3, PPP1R15A, and IL6 among the targets.

## Conclusion

Transcriptomic analyses and validation reveal the importance of the MAP2K1/2/GPAT4-DDIT3 pathway in driving cellular senescence following ATF6α ectopic expression in MCF-7 cells. This study contributes to our understanding of the initial molecular events underlying ER stress-induced cellular senescence in breast cancer cells, providing a foundation for exploring cell type- and stressor-specific responses in cellular senescence induction.

## Introduction

Cellular senescence is a stress response activated by various stimuli, such as telomere attrition, oxidative stress, and oncogene activation. Our previous study has revealed that endoplasmic reticulum (ER) stress plays a role in cellular senescence elicited by NBR1 abrogation, UV irradiation, or ectopic expression of HRAS-V12 [1]. Furthermore, we have discovered that ATF6α, a key unfolded protein response (UPR) sensor molecule, mediates ER stress-induced cellular senescence [1]. The ectopic expression of ATF6α alone was sufficient to cause cellular senescence in MCF-7 human breast cancer cells as well as Caki-1 human renal cancer cells. To unravel the molecular mechanism underlying cellular senescence driven through the ER stress-ATF6α pathway, we conducted transcriptomic analyses on MCF-7 cells expressing ATF6α. MCF-7 cells are used as a human breast cancer cell line model due to its estrogen receptor (ER) (+)/PR(+)/HER2(−) status [2]. Indeed, over 70% of breast cancer patients have been known in the status of estrogen receptor (ER) (+)/PR(+)/HER2(−), thus MCF-7 cells are thought to be one of the relevant cancer cell lines for studying breast cancer [3]. In particular, our study focused on the early stage of cellular senescence, examining transcriptomic changes as early as 6 and 9 hours after ectopic expression of ATF6α.

Cancer therapies elicit cellular senescence in both cancer cells and their microenvironment, a phenomenon known as therapy-induced cellular senescence (TIS). While TIS may be a desired outcome of cancer therapy, senescent cells can inadvertently promote the growth of neighboring cancer cells through the secretion of growth factors, contributing to the development of therapy resistance by enhancing cell survival. Moreover, cellular senescence has been associated with the promotion of epithelial-mesenchymal transition [4], metastasis [5] and generation of cancer stem cells [6]. During cancer therapy, ER stress is known to be induced [7, 8]. Notably, ATF6α pathway of ER stress/UPR enhances cell survival against cancer therapy [7], suggesting that TIS may be mediated through ER stress, particularly via the ATF6α pathway.

Transcriptomics, the study of RNA transcripts produced by the genome, connects the genome, proteome, and cellular phenotype. Recent technological advances in sequencing have enabled the comprehensive profiling of senescence-associated signatures at the genomic level. Various experimental models and environmental contexts have been employed to investigate the transcriptomic and proteomic patterns of senescent cells [9–12]. These studies unveiled the heterogenous nature of cellular senescence, with distinct transcriptome signatures associated with cell types and specific senescence-inducing stresses [11]. Moreover, transcriptomic states of senescent cells evolve over time. Increasing evidence emphasizes the significance of

time-series gene expression data for comprehending the progression of disease and developmental processes [13, 14]. Given the heterogenous and temporally dynamic nature of cellular senescence, it is essential to explore the transcriptomic profile for each cell type exposed to a specific senescence-inducing stimulus at different stages of cellular senescence. In this study, our goal is to comprehend the early-stage molecular mechanism of cellular senescence induced or mediated by the ER stress-ATF6α pathway in the MCF-7 breast cancer cell line.

Consequently, the modulation of cellular senescence in cancer therapy is currently an actively investigated area. Understanding the molecular mechanisms underlying ER stress-induced cellular senescence holds promise for the development of novel strategies in cancer therapy. Our study aims to provide insight into the early-stage mechanisms driving ER stress-induced cellular senescence in MCF-7 human breast cancer cells.

## Methods

### Whole transcriptome sequencing

For the whole transcriptome sequencing of individual MCF-7 cells from three distinct time points (baseline, 6 hours, and 9 hours after ATF6α ectopic expression), total RNA was extracted with Trizol reagent (Invitrogen, USA). Assessment of RNA quality was conducted using the Agilent 2100 bioanalyzer (Agilent Technologies, Amstelveen, NL). RNA quantification was achieved with the ND-2000 Spectrophotometer (Thermo Inc., DE, USA). Libraries were prepared from total RNA utilizing the NEBNext Ultra II Directional RNA-Seq Kit (NEB Inc., UK). mRNA isolation was carried out using the Poly(A) RNA Selection Kit (Lexogen, Inc., Vienna, Austria), and subsequent cDNA synthesis followed the manufacturer's instructions. Indexing was performed using the Illumina indexes 1–12 (Illumina, Inc., USA). The enrichment step involved PCR. Subsequently, libraries were checked using the Agilent 2100 bioanalyzer and the TapeStation HS D1000 Screen Tape (Agilent Technologies, Amstelveen, NL) to evaluate the mean fragment size. Quantification was performed using the library quantification kit on a StepOne Real-Time PCR System (Life Technologies, Inc., USA). High-throughput sequencing employed paired-end 100 sequencing on HiSeq X10 and NovaSeq 6000 platforms (Illumina, Inc., USA).

### Whole transcriptome analysis

Quality assessment of raw sequences was performed by FastQC (v.0.11.9). Low-quality and adapter sequences were eliminated with trimgalore (v.0.6.5–1) before the analysis. The remaining sequences were aligned to the human GRCh38 reference genome using the 2-pass method of STAR (v2.7.3a) [15]. Sequence reads were assigned to genes and the count table was generated by featureCounts (v.2.0.0) [16].

### Differentially expressed genes (DEGs) analysis

Differential expression analysis utilized the DESeq2 R package [17]. For two group comparisons between MCF-7 cells without ATF6α ectopic expression and those after 6 (6h-DEGs) or 9 hours of overexpressing ATF6α (9h-DEGs), the log2 transformed fold change (log2FC) and $p$-values were calculated. False discovery rate (FDR)-adjusted $p$-values were generated using the Benjamini–Hochberg method. DEGs were defined by the following criteria: FDR < 0.05 and |log2FC| > 1. To identify significant changes in time-related gene expression patterns across three-time points (time-related DEGs), the maSigPro R package [18] designed specifically for analyzing time-series gene expression data was employed. Stepwise regression was conducted with an R square of 0.8 and a Benjamini-Hochberg FDR of 0.05.

## Pathway analysis of cellular senescence

In order to unravel the interconnected pathways associated with DEGs, our study employed Ingenuity pathway analysis (IPA) software (Qiagen, Redwood City, CA). The analysis encompassed network analysis, enriched function annotation, upstream regulator analysis, and canonical pathway analysis. IPA utilizes a network generation algorithm to split the network map between molecules into distinct networks and assigns scores to each network. The scores reflect the degree of concordance with the set of focus genes in the Ingenuity analysis. The resulting networks are graphically represented as diagrams illustrating the biological connections and relationships among genes and their corresponding protein products. The consistency score in upstream regulator analysis denotes the extent of agreement between the predicted activation or inhibition of upstream regulators and the expression patterns of its downstream target genes.

## Gene set enrichment analysis (GSEA)

To interpret gene expression data by analyzing the shared common biological functions of gene sets, we employed Gene Set Enrichment Analysis software from the Broad Institute [19]. A pre-ranked GSEA analysis was conducted using the Hallmark gene set collection (50 gene sets) from the Molecular Signature Database (MSigDB v2022.1). According to the established criteria for GSEA, FDR < 0.25 was used as a cutoff value.

## Reverse transcription-polymerase chain reaction (RT-PCR)

Total cellular RNA was isolated using Trizol reagent (Invitrogen, USA) according to the manufacturer's instructions. RT-PCR was carried out as previously described [20] using the following primers: β-actin, 5′-CAGAGCAAAGAGGCATC-3′ (forward) and 5′−GGTAGATGGGCACAGTAT-3′ (reverse); ATF3, 5′-AGGCAATGTACTCTTCCGATGT-3′ (forward) and 5′-ACAGAGGACCTGCCATCATACT-3′ (reverse); ATF6α, 5′-TTTCCGTGACTAAACCTGTCCT-3′ (forward) and 5′-TGTTCCAACATGCTCATAGGTC-3′ (reverse); CHAC1, 5′-TGAATACTTGCTGCGTCTGG-3′ (forward) and 5′-TATCTGCTCAGTGGGCTCAA-3′ (reverse); DDIT3, 5′-CAGATGAAAATGGGGGTACCTA-3′ (forward) and 5′-AATGACCACTCTGTTTCCGTTT-3′ (reverse); EIF2AK3, 5′-AGCCCACCAGTAGCAAATCT-3′ (forward) and 5′-ATCCTGGTCCATTGCAGTCA-3′ (reverse); HERPUD1, 5′-CAGGGACTTGCTTCCAAAGGA-3′ (forward) and 5′-GGTGGAACAAAAGCCCCTGA-3′ (reverse); HSPA5, 5′-TGGACGGTTTCACCACACTT-3′ (forward) and 5′-TTTGCCGAGGGCATCACATA-3′ (reverse).

## Small interfering RNA (siRNA) and plasmid transfection

The plasmid for ATF6α cDNA (p3xFLAG-ATF6) was acquired from Addgene (Watertown, USA). siRNA and plasmid transfections were performed using Lipofectamine RNAiMAX and Lipofectamine™ 2000 (Thermo Fisher Scientific, USA), respectively, according to the manufacturer's instructions. siRNAs were purchased from Bioneer (Korea). The coding strand sequences of the siRNAs were as follows: 5′-CCUACGCCACCAAUUUCGU-3′ (control), MAP2K1-1, 5′-UCUACUGUGGUGAUCUGUA-3′; MAP2K1-2, 5′-CUGUCUACUGUGGUGAUCU-3′; MAP2K2-1, 5′-UCACCAUCAACCCUACCAU-3′; MAP2K2-2, 5′-CUCACAAACCACACCUUCA-3′; PDGFB-1, 5′-CUCGAUCCGCUCCUUUGAU-3′; PDGFB-2, 5′-GUGUACUGCACAAGGACAU-3′; DDIT3-1, 5′-CAGAAGUGGCUACUGACUA-3′; DDIT3-2, 5′-CUGCAAGAGGUCCUGUCUU-3′.

## Senescence associated-β-galactosidase (SA-β-gal) staining

SA-β-gal staining was performed as described by Kim *et al.* [20]. In brief, cells in 60-mm dishes were fixed with 0.2% glutaraldehyde in PBS for 20 min at 25°C. Subsequently, the fixed cells

were incubated in the SA-β-gal staining solution [1 mg/ml 5-bromo- 4-chloro-3-indolyl-β-D-galactopyranoside, 5 mM $K_3Fe(CN)_6$, and 2 mM $MgCl_2$ in PBS, (pH 6.0)], for 16 hours at 37°C. The stained cells were observed using an inverted fluorescence microscope (Eclipse 80i, Nikon, USA).

## Results

### DEGs regulated by ATF6α ectopic expression

To explore the genes exhibiting significant expression changes resulting from ATF6α ectopic expression in MCF-7 cells, we conducted whole transcriptome sequencing across three distinct time points (Fig 1).

At 6 and 9 hours after ATF6α overexpression, we identified a total of 12 and 79 genes of DEGs, referred to as 6h-DEGs and 9h-DEGs, respectively. These genes were selected based on a FDR < 0.05 and a lower fold change (FC) cutoff of |log2FC| > 1 (Tables 1 and 2).

Employing time-series analysis, 20 genes were identified as time-related DEGs (Table 3).

In MCF-7 cells overexpressing ATF6α for 6 hours, genes such as *ATF6*, *DNAJB9*, *CHAC1*, *HERPUD1*, *DDIT3*, and *ATF3* were included in 6h-DEGs (Table 1). Notably, all 6h-DEGs were also detected in 9h-DEGs. In MCF-7 cells overexpressing ATF6α for 9 hours, genes that exhibited differential expression compared to the 6-hour time point included *WIPI1*, *PPP1R15A*, *EIF2AK3*, *GABARAPL1*, *HSPA5*, *PER1*, *MXD1*, *IL6*, and *PIK3R5* (Table 2). The mRNA levels of selected 6h-DEGs (S1A, S9 and S10 Figs) and 9h-DEGs (S1B, S11 and S12 Figs) were validated through RT-PCR analysis. The results show that the majority, though not all, of the indicated DEGs are up-regulated in their mRNA expression in a time-dependent manner following ATF6α overexpression.

### Gene network analysis

To delineate the implicated pathways and investigate the potential signaling networks among the DEGs, we have employed the IPA software. Fig 2 depicts the networks with the highest scores for each DEGs dataset.

The top-scoring network derived from the 6h-DEGs, characterized by *ATF3*, *ATF6*, and *DDIT3*, was related to cell-to-cell signaling and interaction, cellular compromise, and cellular function and maintenance (score = 22, Fig 2A). The top-scoring network for the 9h-DEGs, also consisting of *ATF3*, *ATF6*, and *DDIT3*, was associated with cell-to-cell signaling and inter-action, nutritional disease, and organismal injury and abnormalities (score = 41, Fig 2B). Additional networks from 9h-DEGs were found to be associated with the signaling pathway of cellular senescence (S2 Fig). The network analysis involving time-related DEGs revealed only one network encompassing *ATF3*, *ATF6*, and *TP53* (score = 9, Fig 2C). This network was linked to cancer, cell death and survival, and neurological disease.

### Upstream regulator analysis

Utilizing the regulator effect function of the IPA software, we investigated potential regulators that could explain the differences in expression patterns between MCF-7 cells without overex-pressing ATF6α and those with overexpressing ATF6α for 9 hours. This analysis unveiled six key regulators and eleven target genes (Fig 3 and S3A–S3P Fig).

The identified regulators encompass cyclin dependent kinase 19 (CDK19), platelet-derived growth factor BB (PDGFBB), tumor necrosis factor (TNF), RB transcriptional corepressor 1 (RB1), mitogen-activated protein kinase kinase 1/2 (MAP2K1/2), and glycerol-3-phosphate acyltransferase 4 (GPAT4). Notably, CDK19, PDGFBB, TNF, and RB1 are linked to cytostasis,

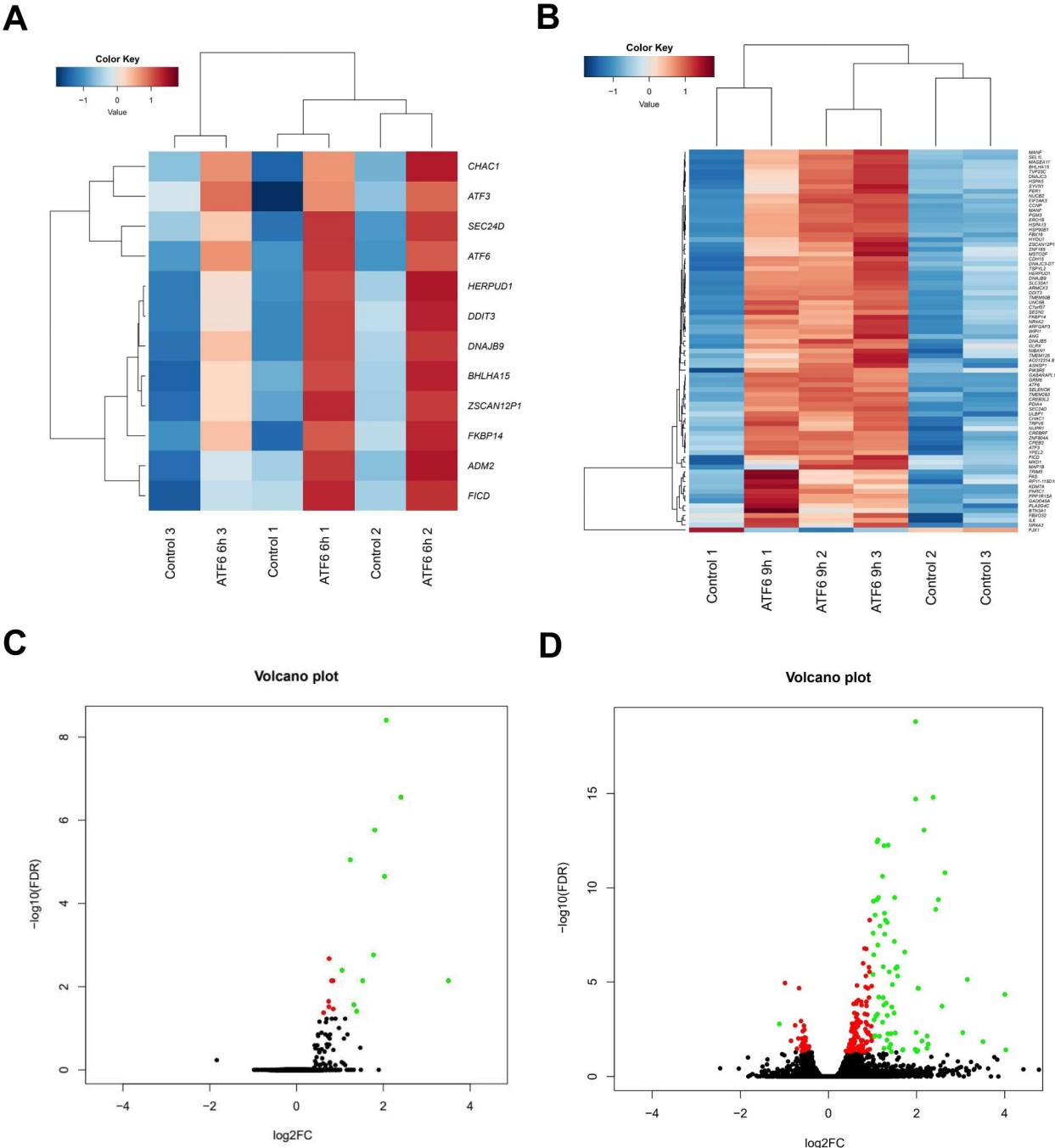

**Fig 1. Heatmap and volcano plot illustrating differential gene expression analysis following ectopic expression of ATF6α in MCF-7 cells.** Differential gene expression analysis was performed between MCF-7 cells without ATF6α ectopic expression and those ectopically expressing ATF6α for 6 or 9 hours. MCF-7 cells were transfected with ATF6α cDNA (MCF-7/ATF6 OE) or an empty vector DNA as a control (MCF-7/Control). (A) 6h-DEGs Heatmap: Heatmap depicting 12 DEGs with FDR < 0.05 and |log2FC| > 1 in MCF-7 cells at 6 hours after ectopic expression of ATF6α. Red indicates high expression, and blue indicates low expression. (B) 9h-DEGs Heatmap: Heatmap illustrating 79 DEGs with FDR < 0.05 and |log2FC| > 1 in MCF-7 cells at 9 hours after ATF6α ectopic expression. (C) Volcano plot (6h-DEGs): Distribution of 6h-DEGs between MCF-7/Control and MCF-7/ATF6 OE cells. The X-axis represents the log2FC of individual genes, and the Y-axis represents the negative logarithm of their FDR to base 10 [−log10(FDR)]. Black dots: FDR > 0.05; Red dots: FDR ≤ 0.05 and |log2FC| ≤ 1; Green dots: 6h-DEGs with FDR ≤ 0.05 and |log2FC| > 1. ATF6α is excluded from the plot due to space limitations (−log10FDR = 52.89, log2FC = 3.23). (D) Volcano plot (9h-DEGs): Distribution of 9h-DEGs between MCF-7/Control and MCF-7/ATF6 OE cells. ATF6α is excluded due to space limitations (−log10FDR = 271.60, log2FC = 4.87).

**Table 1. DEGs in MCF-7 cells at 6 hours after ectopic expression of ATF6α.**

| Genes | log2FC | p-value | FDR | Function |
|---|---|---|---|---|
| ATF6 | 3.23 | 9.82E-58 | 1.29E-53 | ER stress |
| DNAJB9 | 2.07 | 5.96E-13 | 3.93E-09 | Heat Shock Protein |
| CHAC1 | 2.41 | 6.33E-11 | 2.78E-07 | Unfolded Protein Response |
| HERPUD1 | 1.80 | 5.21E-10 | 1.71E-06 | ER-associated degradation |
| SEC24D | 1.24 | 3.38E-09 | 8.91E-06 | Vesicle trafficking |
| DDIT3 | 2.03 | 1.02E-08 | 2.24E-05 | Apoptosis |
| ATF3 | 1.78 | 9.18E-07 | 1.73E-03 | Cellular stress response |
| FKBP14 | 1.05 | 2.75E-06 | 4.02E-03 | Protein folding |
| ZSCAN12P1 | 1.53 | 6.68E-06 | 7.21E-03 | Pseudogene |
| BHLHA15 | 3.50 | 7.11E-06 | 7.21E-03 | Unfolded Protein Response |
| ADM2 | 1.32 | 3.11E-05 | 2.73E-02 | Peptide hormone |
| FICD | 1.39 | 5.32E-05 | 3.90E-02 | ER stress |

log2FC, log 2 fold change; FDR, false discovery rate (FDR < 0.05 and |log2FC| > 1)

while MAP2K1/2 and GPAT4 are associated with cell viability and cell death of connective tissue cells.

## Enriched functional annotation and canonical pathway analysis

To discern the functional implications of the three types of DEGs, enriched functional annotation was conducted, identifying significantly enriched biological functions. The top 10 functions along their corresponding p-values are listed in Table 4 (FDR < 0.05, |log2FC| > 1).

Commonly enriched functional annotations in both 6h-DEGs and 9h-DEGs encompass endoplasmic reticulum stress response, asbestosis, and stress response of cells, as well as functions associated with cell death and apoptosis.

Canonical pathway analysis revealed shared signaling pathways across all three types of DEGs. These included UPR, ER stress pathway, and endocannabinoid cancer inhibition pathway (Table 5).

Notably, genes such as ATF6, ATF3, and DDIT3 were recurrently emerged in these pathways. In addition, canonical pathways derived from both 6h-DEGs and 9h-DEGs included NRF2-mediated oxidative stress response and EIF2 signaling, while those from time-related DEGs encompassed PI3K signaling in B lymphocytes, ID1 signaling pathway and sirtuin signaling pathway (Table 5). These results suggest that the ectopic expression of ATF6α in breast cancer cells trigger diverse signaling pathways that impact the fate of these cells, including those associated with proliferation, cell death, or senescence.

## GSEA

To further investigate molecular signaling pathways that are differentially activated in MCF-7 cells overexpressing ATF6α for 6 or 9 hours, GSEA was conducted. The analysis revealed significant differences (FDR < 0.25) in the enrichment of the hallmark gene sets. The top 5 enriched pathways at each time points were selected based on their normalized enriched scores (NES). As shown in Fig 4A, the Myc targets v1 pathway, mTORC1 signaling pathway, UPR pathway, androgen response pathway, and G2M checkpoint pathway were differentially enriched in MCF-7 cells after 6 hours of ATF6α ectopic expression.

Additionally, the interferon alpha response pathway, interferon gamma response pathway, IL6-JAK/STAT3 signaling pathway, UPR pathway, and KRAS signaling pathway were

**Table 2. DEGs in MCF-7 cells at 9 hours after ectopic expression of ATF6α.**

| Genes | log2FC | p-value | FDR | Function |
|---|---|---|---|---|
| ATF6 | 4.87 | 1.50E-276 | 2.53E-272 | ER stress |
| SEC24D | 1.98 | 1.80E-23 | 1.52E-19 | Vesicle trafficking |
| DNAJB9 | 2.38 | 2.81E-19 | 1.58E-15 | Heat Shock Protein |
| HERPUD1 | 1.98 | 4.63E-19 | 1.95E-15 | ER-associated degradation |
| DDIT3 | 2.17 | 2.58E-17 | 8.71E-14 | Apoptosis |
| PDIA4 | 1.12 | 1.04E-16 | 2.92E-13 | Protein folding |
| CREB3L2 | 1.11 | 1.49E-16 | 3.59E-13 | Unfolded Protein Response |
| ARMCX3 | 1.35 | 2.60E-16 | 5.48E-13 | Tumor suppression |
| HSP90B1 | 1.26 | 3.14E-16 | 5.88E-13 | Heat Shock Protein |
| ATF3 | 2.64 | 9.49E-15 | 1.60E-11 | Cellular stress response |
| PGM3 | 1.23 | 1.60E-14 | 2.46E-11 | Glycogen synthesis |
| WIPI1 | 1.50 | 2.43E-13 | 3.31E-10 | Protein assembly |
| MANF | 1.13 | 2.55E-13 | 3.31E-10 | Cell proliferation and death |
| CHAC1 | 2.49 | 3.66E-13 | 4.23E-10 | Unfolded Protein Response |
| HSPA13 | 1.10 | 3.77E-13 | 4.23E-10 | Heat Shock Protein |
| TMEM263 | 1.02 | 4.87E-13 | 5.13E-10 | Unknown |
| ERO1B | 2.44 | 1.40E-12 | 1.39E-09 | Protein folding |
| HYOU1 | 1.27 | 2.38E-12 | 2.23E-09 | Heat Shock Protein |
| TMEM50B | 1.06 | 3.14E-12 | 2.79E-09 | Unknown |
| NUCB2 | 1.29 | 6.12E-12 | 5.16E-09 | Calcium homeostasis |
| PPP1R15A | 1.33 | 9.02E-12 | 6.91E-09 | Cellular stress response |
| EIF2AK3 | 1.17 | 1.45E-11 | 1.06E-08 | Unfolded Protein Response |
| SELENOK | 1.01 | 3.64E-11 | 2.56E-08 | The Selenoprotein gene |
| ARFGAP3 | 1.28 | 4.31E-11 | 2.90E-08 | GTPase-activating protein |
| GABARAPL1 | 1.50 | 1.09E-10 | 7.05E-08 | Autophagosome assembly |
| PNRC1 | 1.12 | 1.76E-10 | 1.10E-07 | Unknown |
| HSPA5 | 1.73 | 4.59E-10 | 2.58E-07 | Heat Shock Protein |
| SEL1L | 1.03 | 6.66E-10 | 3.62E-07 | Protein degradation |
| ADM2 | 1.56 | 3.13E-09 | 1.55E-06 | Peptide hormone |
| SESN2 | 1.24 | 3.11E-09 | 1.55E-06 | Cell growth and survival |
| FBXO16 | 1.56 | 3.49E-09 | 1.64E-06 | Protein degradation |
| GADD45A | 1.53 | 4.12E-09 | 1.87E-06 | Cellular stress response |
| FKBP14 | 1.38 | 6.70E-09 | 2.90E-06 | Protein folding |
| SLC33A1 | 1.01 | 9.31E-09 | 3.92E-06 | Acetylation-mediated secretion |
| TSPYL2 | 1.57 | 1.20E-08 | 4.81E-06 | Chromatin remodeling |
| CDH15 | 3.15 | 1.88E-08 | 7.37E-06 | Calcium-dependent cell adhesion |
| DNAJC3 | 1.45 | 3.66E-08 | 1.37E-05 | Heat Shock Protein |
| TRPV6 | 2.03 | 6.31E-08 | 2.13E-05 | Calcium channel |
| ZSCAN12P1 | 2.04 | 6.73E-08 | 2.18E-05 | Pseudogene |
| BHLHA15 | 4.00 | 1.43E-07 | 4.54E-05 | Unfolded Protein Response |
| PER1 | 1.14 | 2.01E-07 | 6.28E-05 | Circadian rhythms |
| CPEB3 | 1.24 | 2.19E-07 | 6.73E-05 | Translation regulation |
| NR4A2 | 1.31 | 4.74E-07 | 1.31E-04 | Nuclear receptor |
| YPEL2 | 1.26 | 6.47E-07 | 1.68E-04 | Unknown |
| C7orf57 | 5.84 | 7.13E-07 | 1.82E-04 | Unknown |
| NIBAN1 | 2.58 | 7.72E-07 | 1.91E-04 | Apoptosis |
| ZNF165 | 1.44 | 8.66E-07 | 2.12E-04 | Transcription regulation |

*(Continued)*

**Table 2.** (Continued)

| Genes | log2FC | *p*-value | FDR | Function |
|---|---|---|---|---|
| *TRIM5* | 1.49 | 1.85E-06 | 4.39E-04 | Protein degradation |
| *CREBRF* | 1.11 | 2.18E-06 | 5.04E-04 | Unfolded Protein Response |
| *FICD* | 1.33 | 2.66E-06 | 5.90E-04 | Protein adenylation |
| *UNC5B* | 1.08 | 2.93E-06 | 6.41E-04 | Apoptosis |
| *SYVN1* | 1.03 | 4.87E-06 | 1.03E-03 | ER-associated degradation |
| *TMEM125* | 1.23 | 6.79E-06 | 1.35E-03 | Unknown |
| *FJX1* | -1.11 | 8.73E-06 | 1.67E-03 | Unknown |
| *NR4A3* | 1.99 | 3.02E-05 | 4.71E-03 | Nuclear receptor |
| *ZNF804A* | 3.05 | 3.14E-05 | 4.81E-03 | Transcription regulation |
| *NUPR1* | 1.52 | 3.30E-05 | 5.01E-03 | Nuclear protein |
| *MSTO2P* | 1.42 | 3.72E-05 | 5.45E-03 | Pseudogene |
| *TVP23C* | 1.35 | 3.71E-05 | 5.45E-03 | Vesicle trafficking |
| *DNAJB5* | 1.05 | 4.91E-05 | 7.07E-03 | Heat Shock Protein |
| *ENSG00000251095* | 2.24 | 5.08E-05 | 7.20E-03 | Unknown |
| *MXD1* | 1.16 | 5.26E-05 | 7.39E-03 | Cell proliferation and apoptosis |
| *KDM7A* | 1.04 | 5.57E-05 | 7.69E-03 | Histone demethylation |
| *FAS* | 1.33 | 9.31E-05 | 1.19E-02 | Apoptosis |
| *ENSG00000237017* | 2.12 | 1.10E-04 | 1.33E-02 | Unknown |
| *ASNSP1* | 3.51 | 1.23E-04 | 1.44E-02 | Pseudogene |
| *BTN3A1* | 2.26 | 1.72E-04 | 1.93E-02 | The MHC gene |
| *GRM8* | 5.80 | 1.76E-04 | 1.94E-02 | Glutamate receptor |
| *MAP1B* | 1.41 | 1.82E-04 | 2.00E-02 | Microtubule assembly |
| *GLRX* | 1.38 | 2.92E-04 | 2.97E-02 | Glutaredoxin |
| *IL6* | 2.24 | 3.33E-04 | 3.25E-02 | Interleukin-6 |
| *DNAJC3-DT* | 1.26 | 3.43E-04 | 3.28E-02 | lncRNA |
| *FBXO32* | 1.97 | 3.98E-04 | 3.67E-02 | Protein degradation |
| *CCNP* | 1.69 | 4.16E-04 | 3.77E-02 | Cell cycle |
| *MAGEA11* | 4.02 | 4.50E-04 | 3.95E-02 | The MAGEA gene |
| *PLA2G4C* | 1.68 | 4.63E-04 | 4.03E-02 | Phospholipase A2 |
| *PIK3R5* | 2.03 | 4.87E-04 | 4.21E-02 | Cell proliferation and differentiation |
| *ANG* | 2.01 | 5.95E-04 | 4.84E-02 | Angiogenesis |
| *ULBP1* | 1.43 | 6.10E-04 | 4.92E-02 | The ligand of NKG2D |

log2FC, log 2 fold change; FDR, false discovery rate (FDR < 0.05 and |log2FC| > 1)

enriched in MCF-7 cells after 9 hours of ATF6α ectopic expression (Fig 4B). These findings suggest that the Myc and mTORC1 signaling pathways, previously associated with cellular senescence, are among the earliest activated signaling pathways contributing to cellular senescence induction by ectopic expression of ATF6α.

## Validation of regulatory molecules and pathways identified through upstream regulator analysis

To validate the regulatory molecules and pathways identified through upstream regulator analysis, we conducted functional experiments targeting four regulators, namely PDGF-BB, TNF, MAP2K1/2 and GPAT4, predicted to exhibit increased expression or activity. Using siRNA-mediated knockdown, we individually assessed the impact of these regulators on cellular

**Table 3. DEGs in MCF-7 cells across three time points.**

| Genes | p-value | FDR | Function |
|---|---|---|---|
| MTCO1P17 | 4.43E-07 | 1.80E-03 | Pseudogene |
| IMPA1P1 | 4.43E-07 | 1.80E-03 | Pseudogene |
| PKHD1L1 | 4.43E-07 | 1.80E-03 | Polycystic kidney and hepatic diseases |
| LINC02373 | 4.43E-07 | 1.80E-03 | lncRNA |
| Y_RNA | 4.43E-07 | 1.80E-03 | The RNA gene |
| GOLGA6A | 4.43E-07 | 1.80E-03 | The PML gene |
| MRPL15P1 | 4.43E-07 | 1.80E-03 | Pseudogene |
| ENSG00000283405 | 4.43E-07 | 1.80E-03 | Unknown |
| PPIF | 1.22E-05 | 4.40E-02 | Protein folding |
| ATF6 | 1.82E-05 | 4.75E-02 | ER stress |
| ATF3 | 2.84E-05 | 4.75E-02 | Cellular stress response |
| SCTR | 2.92E-05 | 4.75E-02 | Pancreatic cancer and autism |
| ARF4 | 2.49E-05 | 4.75E-02 | Vesicular trafficking |
| ENSG00000234686 | 2.92E-05 | 4.75E-02 | Unknown |
| GULOP | 2.92E-05 | 4.75E-02 | Pseudogene |
| ENSG00000232283 | 2.92E-05 | 4.75E-02 | Unknown |
| ENSG00000255367 | 2.92E-05 | 4.75E-02 | Unknown |
| GABARAPL1 | 2.76E-05 | 4.75E-02 | Autophagosome assembly |
| ENSG00000224647 | 2.92E-05 | 4.75E-02 | Unknown |
| B3GALT5-AS1 | 2.92E-05 | 4.75E-02 | lncRNA |

log2FC, log 2 fold change; FDR, false discovery rate (FDR < 0.05 and |log2FC| > 1)

senescence, measured by senescence-associated β-galactosidase (SA-β-gal) staining (Fig 5 and S3A and S3B Fig).

MCF-7 cells were co-transfected with ATF6α cDNA and siRNA targeting one of the regulatory genes, and after 5 days, cellular senescence was evaluated. Remarkably, knockdown of PDGFB, MAP2K1/2 (MAP2K1 or MAP2K2) or GPAT4 significantly mitigated cellular senescence (Fig 5A–5F).

Subsequently, we explored the roles of downstream targets of PDGF-BB, MAP2K1/2 and GPAT4 in senescence induction. Among the annotated downstream targets of MAP2K1/2 and GPAT4, knockdown of DDIT3 exhibited a preventive effect on cellular senescence (Fig 5G and 5H). Furthermore, MAP2K1/2 and GPAT4 were observed to upregulate DDIT3 mRNA expression in that knockdown of MAP2K1, MAP2K2 or GPAT4 inhibited the increase in DDIT3 mRNA expression induced by the ectopic expression of ATF6α (Fig 6 and S6–S8 Figs).

Taken together, these findings suggest that the MAP2K1/2/GPAT4-DDIT3 axis may play a pivotal role in cellular senescence induced by the ectopic expression of ATF6α.

Similarly, among the annotated targets of PDGF-BB, knockdown of PPP1R15A and IL-6 attenuated cellular senescence (S3E and S3F Fig). However, RT-PCR analysis revealed that knockdown of PDGFB did not distinctly affect the mRNA expression of PPP1R15A and IL6, indicating that PDGF-BB may not directly regulate their mRNA expression (S4 and S13 Figs). This finding emphasizes the necessity for further exploration to better understand the roles and interplay between PDGF-BB and PPP1R15A/IL6 in the context of cellular senescence.

ATF3 was identified as a target of both MAP2K1/2 and GPAT4, frequently appearing in the gene network analysis and enriched functional annotations. However, knockdown of ATF3 did not affect cellular senescence induced by the ectopic expression of ATF6α (S3M and S3N

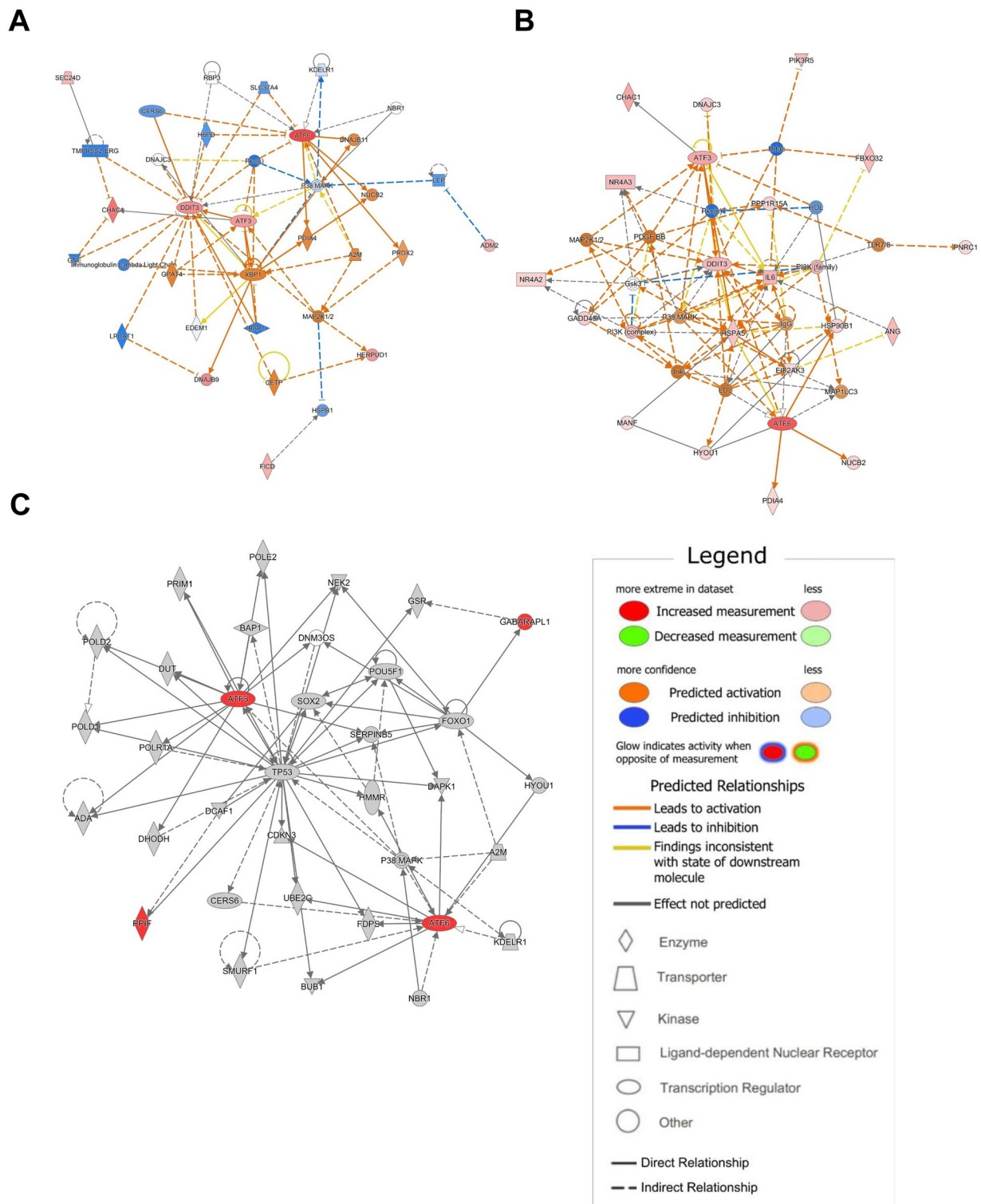

**Fig 2. Gene interaction networks revealed by IPA software.** Utilizing IPA software, network analysis was performed on three distinct sets of DEGs, resulting in top-scoring networks. The legend illustrates the relationship between molecules within each network and their activation states. (A) A gene network of 6h-DEGs with a score of 22, associated with cell-to-cell signaling and interaction, cellular compromise, and cellular function and maintenance. (B) A gene network of 9h-DEGs with a score of 41, associated with cell-to-cell signaling and interaction, nutritional disease, and organismal injury and abnormalities. (C) A gene network of time-related DEGs with a score of 9, associated with cancer, cell death and survival, and neurological disease.

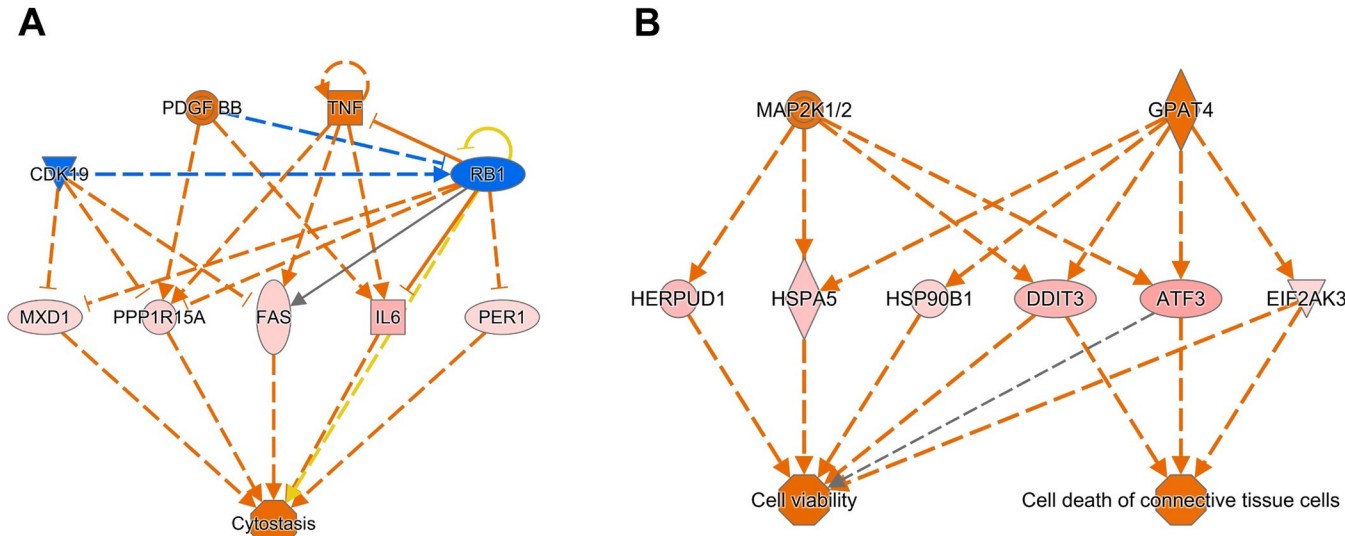

**Fig 3. Networks representing the regulatory effects of 9h-DEGs.** Two potential networks regulating phenotypic or functional outcomes were identified. Nodes at the top indicate upstream regulators, while the nodes in the middle represent genes modulated by these regulators. Nodes at the bottom indicate expected phenotypic consequences resulting from changes in gene expression. (A) Regulatory network associated with cytostasis, featuring a consistency score of 5.37. This network comprises four major upstream regulators: CDK19, PDGF-BB, TNF, and RB1. (B) Regulatory network associated with cell viability and cell death of connective tissue cells, with a consistency score of 4.90. This network involves two major upstream regulators: MAP2K1/2 and GPAT4.

Fig). This suggests a nuanced relationship between ATF3 and cellular senescence, highlighting the complexity of the regulatory network involved in cellular senescence.

Since MCF-7 cells have estrogen receptor (ER)(+)/PR(+)/HER2(-) phenotypes, which represent more than 70% of human breast cancers [3]. We analyzed overall survival (OS) of breast cancer patients with high and low expressions of ATF6α using Kaplan-Meier plotter (https://kmplot.com/analysis), showing that breast cancer patients with high ATF6α expression levels have decreased OS rates compared to those of low ATF6α expression levels. Furthermore, breast cancer patients with high expression levels of the ATF6α-MAP2K1/2-DDIT3 axis also showed decreased OS rates (S5 Fig). The data indicate that cellular senescence might be a key event associated with chemoresistance, epithelial mesenchymal transition (EMT), metastasis and production of cancer stem cell (CSC).

Overall, these results underscore the significance of the MAP2K1/2/GPAT4-DDIT3 axis in the induction of cellular senescence through the ectopic expression of ATF6α in MCF-7 cells.

## Discussion

In this study, we conducted transcriptomic analysis of MCF-7 cells to unravel the intricate molecular mechanisms driving ER stress-induced cellular senescence. Based on our prior discovery that the ER stress-induced senescence is mediated by the UPR sensor molecule ATF6α [1], we induced cellular senescence through the ectopic expression of ATF6α in MCF-7 cells. To investigate the initial molecular events triggering cellular senescence, transcriptomic analysis was conducted at early time points, specifically 6 and 9 hours following ATF6α ectopic expression. Employing 6h-DEGs, 9h-DEGs, and time-related-DEGs, we conducted a comprehensive analysis including network analysis, enriched functional annotation, canonical pathway analysis, GSEA, and upstream regulator analysis.

Validation of the molecules identified in the upstream regulator analysis revealed the significance of the MAP2K1/2-DDIT3 axis in the induction of cellular senescence via ectopic

**Table 4. Top 10 enriched functional annotations of three types of DEGs.**

**6h-DEGs**

| Functional Annotations | *p*-value | Genes |
|---|---|---|
| Endoplasmic reticulum stress response | 2.57E-07 | ATF6, DDIT3, FICD, HERPUD1 |
| Asbestosis | 2.42E-06 | ATF6, DDIT3 |
| Stress response of cells | 9.85E-06 | ATF6, DDIT3, FICD |
| Endoplasmic reticulum stress response of cervical cancer cell lines | 1.09E-05 | ATF6, DDIT3 |
| Cell death of connective tissue cells | 1.57E-04 | ATF3, ATF6, DDIT3 |
| Apoptosis of kidney cancer cell lines | 3.67E-04 | ATF3, DDIT3 |
| Quantity of M2 macrophages | 5.16E-04 | ADM2 |
| Achromatopsia 7 | 5.16E-04 | ATF6 |
| Ehlers-Danlos syndrome with progressive kyphoscoliosis, myopathy, and hearing loss | 5.16E-04 | FKBP14 |
| Cole-Carpenter syndrome type 2 | 5.16E-04 | SEC24D |

**9h-DEGs**

| Functional Annotations | *p*-value | Genes |
|---|---|---|
| Endoplasmic reticulum stress response of cervical cancer cell lines | 8.81E-11 | ATF6, DDIT3, EIF2AK3, HSP90B1, HSPA5 |
| Endoplasmic reticulum stress response | 1.58E-10 | ATF6, DDIT3, EIF2AK3, FICD, HERPUD1, HSP90B1, HSPA5, HYOU1, PPP1R15A |
| Cell death of neuroblastoma cell lines | 1.62E-08 | ANG, CREB3L2, DDIT3, EIF2AK3, FAS, GLRX, HSPA5, IL6, SYVN1 |
| Cell death of pancreas | 5.62E-08 | ATF3, FAS, IL6, MANF, NR4A3 |
| Asbestosis | 3.52E-07 | ATF6, DDIT3, HSPA5 |
| Stress response of cells | 3.90E-07 | ATF6, DDIT3, EIF2AK3, FICD, HSP90B1, HSPA5 |
| Apoptosis of pancreatic cells | 1.14E-06 | FAS, IL6, MANF, NR4A3 |
| Apoptosis of islets of Langerhans | 1.87E-06 | ATF3, FAS, MANF, NR4A3 |
| Contact growth inhibition of colorectal cancer cell lines | 9.87E-06 | HSPA5, MXD1, PPP1R15A |
| Apoptosis of hepatoma cell lines | 1.08E-05 | ATF6, DDIT3, EIF2AK3, FAS, HSPA5, IL6, PPP1R15A |

**time-related DEGs**

| Functional Annotations | p-value | Genes |
|---|---|---|
| Achromatopsia 7 | 4.23E-04 | ATF6 |
| Nodal diffuse large B-cell lymphoma | 8.45E-04 | ATF3 |
| Scurvy | 1.27E-03 | GULOP |
| Asbestosis | 2.11E-03 | ATF6 |
| Cell viability of skin cancer cell lines | 2.53E-03 | ATF6 |
| Cell survival of leukemia cell lines | 2.53E-03 | ATF3 |
| Foveal hypoplasia | 2.53E-03 | ATF6 |
| Size of autophagosomes | 2.96E-03 | GABARAPL1 |
| Cell death of connective tissue cells | 3.43E-03 | ATF3, ATF6 |
| Permeability of mitochondrial membrane | 3.80E-03 | PPIF |

Note: Through differentially expressed gene (DEG) analysis, we identified genes that showed significant expression pattern change between MCF-7 cells without overexpressing ATF6α and MCF-7 cells at 6 or 9 hours after overexpressing of ATF6α (6h-DEGs and 9h-DEGs, respectively) and across three-time points (time-related DEGs). Ingenuity pathway analysis (IPA) software was used to analyze enriched functional annotations with an input of three types of DEGs (FDR < 0.05 and |log2FC| > 1).

**Table 5. Top 10 canonical pathways of three types of DEGs.**

| 6h-DEGs | | | |
|---|---|---|---|
| **Ingenuity Canonical Pathways** | **−log($p$-value)** | **Ratio** | **Genes** |
| Unfolded protein response | 3.53 | 0.03 | *ATF6, DDIT3, DNAJB9* |
| Endoplasmic Reticulum Stress Pathway | 3.18 | 0.10 | *ATF6, DDIT3* |
| ID1 Signaling Pathway | 2.97 | 0.02 | *ATF3, ATF6, BHLHA15* |
| PI3K Signaling in B Lymphocytes | 1.89 | 0.01 | *ATF3, ATF6* |
| Endocannabinoid Cancer Inhibition Pathway | 1.89 | 0.01 | *ATF3, DDIT3* |
| NRF2-mediated Oxidative Stress Response | 1.72 | 0.01 | *DNAJB9, HERPUD1* |
| EIF2 Signaling | 1.72 | 0.01 | *ATF3, DDIT3* |
| γ-glutamyl Cycle | 1.66 | 0.08 | *CHAC1* |
| tRNA Splicing | 1.22 | 0.02 | *FICD* |
| Apelin Pancreas Signaling Pathway | 1.22 | 0.02 | *DDIT3* |
| **9h-DEGs** | | | |
| **Ingenuity Canonical Pathways** | **−log($p$-value)** | **Ratio** | **Genes** |
| Unfolded protein response | 13.4 | 0.14 | *ATF6, DDIT3, DNAJB5, DNAJB9, DNAJC3, EIF2AK3, ERO1B, HSP90B1, HSPA5, PPP1R15A, SEL1L, SYVN1* |
| Endoplasmic Reticulum Stress Pathway | 8.06 | 0.29 | *ATF6, DDIT3, DNAJC3, EIF2AK3, HSP90B1, HSPA5* |
| Aldosterone Signaling in Epithelial Cells | 3.97 | 0.04 | *DNAJB5, DNAJB9, DNAJC3, HSP90B1, HSPA13, HSPA5, PIK3R5* |
| NRF2-mediated Oxidative Stress Response | 3.3 | 0.03 | *DNAJB5, DNAJB9, DNAJC3, EIF2AK3, HERPUD1, HSP90B1, PIK3R5* |
| EIF2 Signaling | 2.46 | 0.03 | *ATF3, DDIT3, EIF2AK3, HSPA5, PIK3R5, PPP1R15A* |
| Autophagy | 2.46 | 0.03 | *CREB3L2, DDIT3, EIF2AK3, GABARAPL1, PIK3R5, WIPI1* |
| Role of PKR in Interferon Induction and Antiviral Response | 2.46 | 0.04 | *ATF3, DNAJC3, FAS, HSP90B1, HSPA5* |
| Endocannabinoid Cancer Inhibition Pathway | 2.31 | 0.03 | *ATF3, CREB3L2, DDIT3, NUPR1, PIK3R5* |
| Protein Ubiquitination Pathway | 2.02 | 0.02 | *DNAJB5, DNAJB9, DNAJC3, HSP90B1, HSPA13, HSPA5* |
| Apelin Pancreas Signaling Pathway | 1.89 | 0.07 | *DDIT3, EIF2AK3, PIK3R5* |
| **time-related DEGs** | | | |
| **Ingenuity Canonical Pathways** | **−log($p$-value)** | **Ratio** | **Genes** |
| PI3K Signaling in B Lymphocytes | 1.54 | 0.01 | *ATF3,ATF6* |
| ID1 Signaling Pathway | 1.52 | 0.01 | *ATF3,ATF6* |
| Sirtuin Signaling Pathway | 1.40 | 0.01 | *GABARAPL1,PPIF* |
| Endoplasmic Reticulum Stress Pathway | 1.35 | 0.05 | *ATF6* |
| Unfolded protein response | 0.86 | 0.01 | *ATF6* |
| Ferroptosis Signaling Pathway | 0.86 | 0.01 | *ARF4* |
| Role of PKR in Interferon Induction and Antiviral Response | 0.86 | 0.01 | *ATF3* |
| Endocannabinoid Cancer Inhibition Pathway | 0.86 | 0.01 | *ATF3* |
| Necroptosis Signaling Pathway | 0.86 | 0.01 | *PPIF* |
| Integrin Signaling | 0.86 | 0.00 | *ARF4* |

Note: Through differentially expressed gene analysis, we identified genes that showed significant expression pattern change between MCF-7 cells without overexpressing ATF6α and MCF-7 cells after 6 or 9 hours of overexpressing ATF6α (6h-DEGs and 9h-DEGs, respectively) and across three-time points (time-related DEGs).

Canonical pathways are identified with an input of three types of DEGs (FDR < 0.05 and |log2FC| > 1) using Ingenuity pathway analysis (IPA) software.

The Fisher exact test was utilized to compute the $p$-value in order to assess whether the correlation between the DEGs and canonical pathway can be attributed to random chance.

The ratio shows how many molecules from the top 100 genes are involved in the specific pathway compared to the total number of molecules involved in the entire canonical pathway.

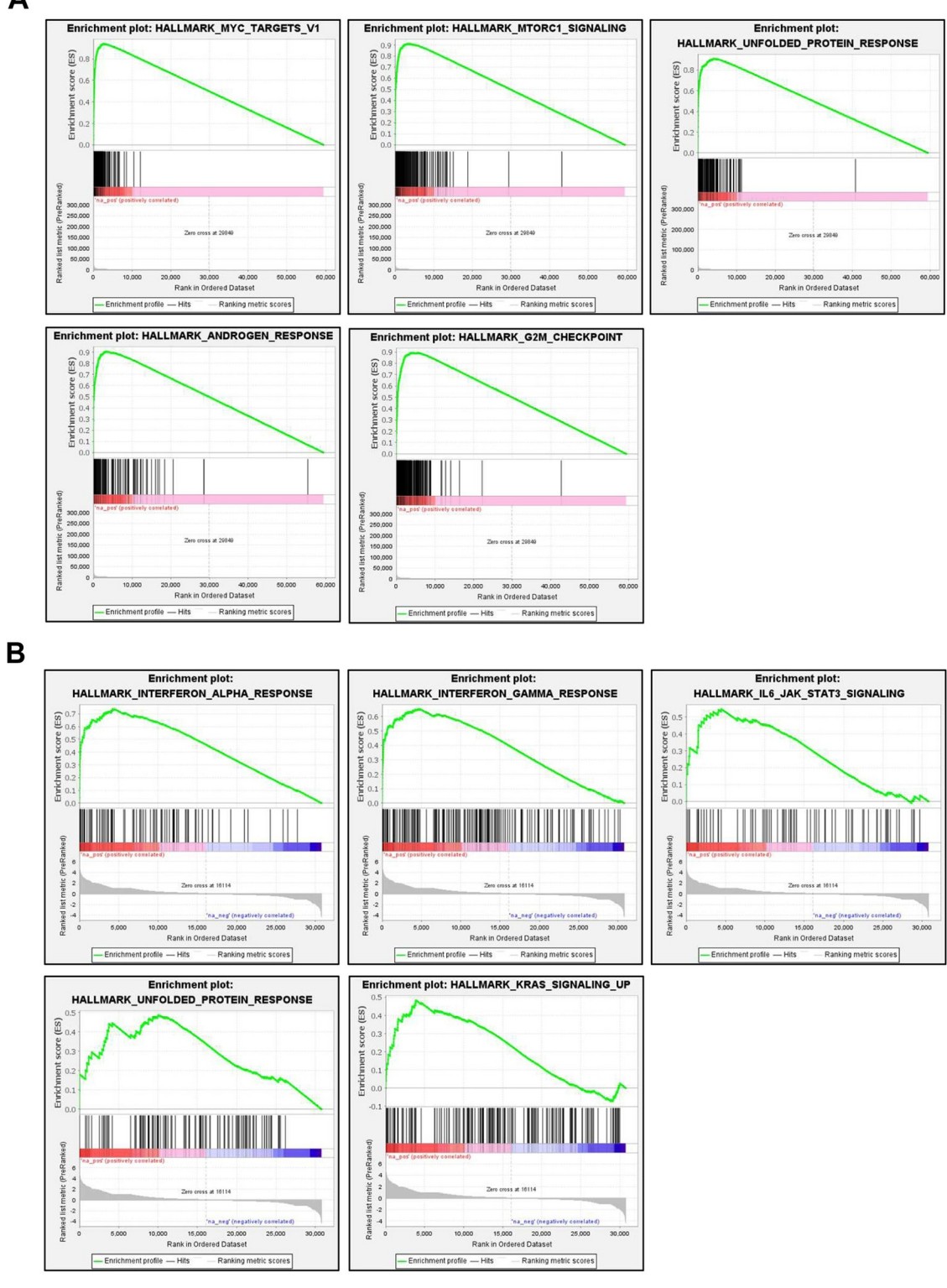

**Fig 4. GSEA using the hallmark gene set collection from the MSigDB.** (A) Top 5 enrichment pathway plots of MCF-7 cells at 6 hours after ATF6α ectopic expression (FDR < 0.25). Differentially enriched pathways include Myc targets, mTORC1 signaling, UPR, androgen response, and G2M checkpoint pathways. (B) Top 5 enrichment pathway plots of MCF-7 cells at 9 hours after ATF6α ectopic expression (FDR < 0.25) (FDR < 0.25). Enriched pathways comprise the interferon alpha response, interferon gamma response, IL6- JAK/STAT3 signaling, UPR, and KRAS signaling pathways.

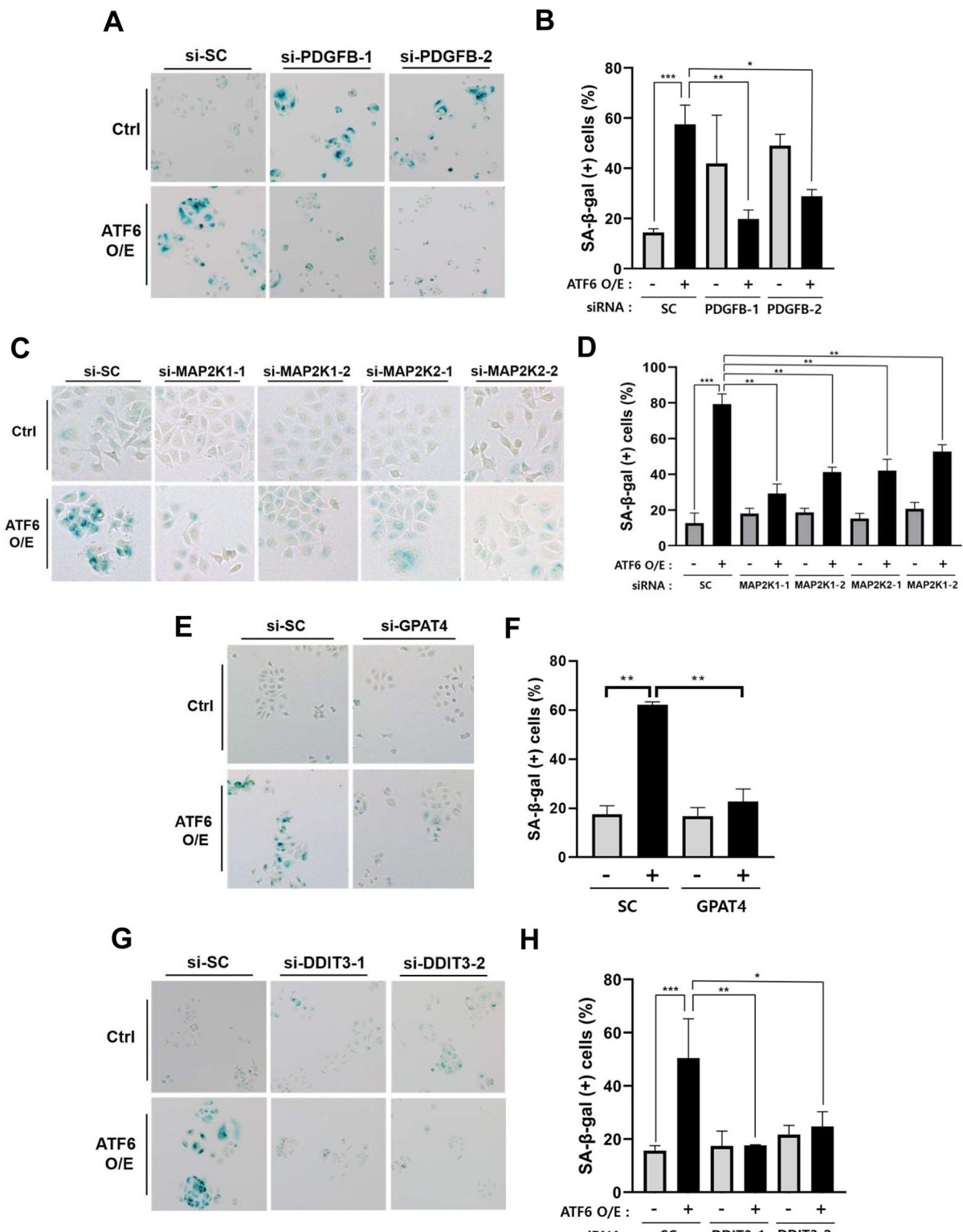

**Fig 5. Validation of roles of PDGFB, MAP2K1/2, GPAT4, and DDIT3 in cellular senescence induced by ATF6α ectopic expression.**
MCF-7 cells were co-transfected with ATF6α cDNA and control siRNA (si-SC) or siRNA targeting the specified genes. Two different siRNAs were employed for each gene. After 5 days, SA-β-gal staining was performed. (A, C, E, G) Light microscopic images represent three independent experiments. (B, D, F, H) Percentages of SA-β-gal (+) cells were plotted. Data represent the mean of triplicate determinations ± standard deviation (S.D.). Ctrl, control; ATF6 O/E, ATF6α overexpression. * $P<0.05$; ** $P<0.01$; *** $P<0.001$.

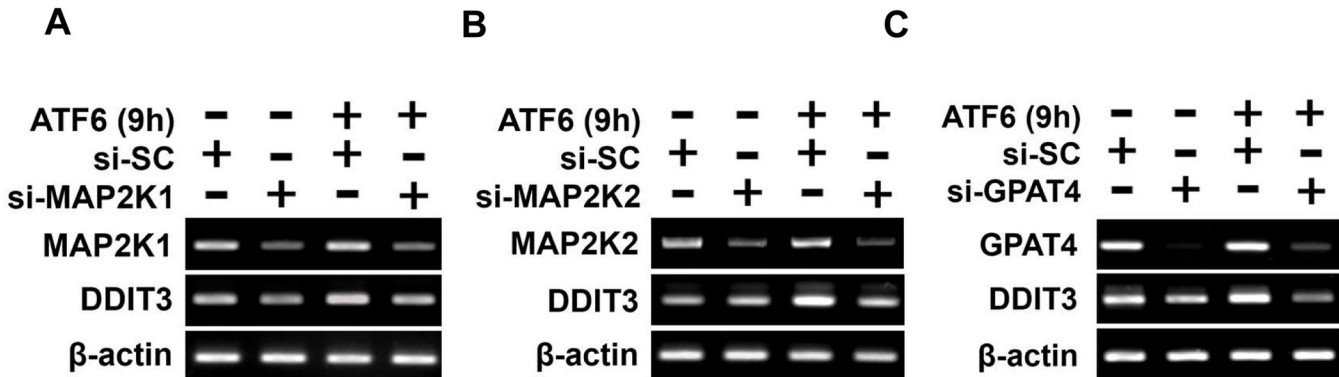

**Fig 6. Regulation of DDIT3 mRNA expression by MAP2K1/2 and GPAT4 9 hours after ATF6α ectopic expression in MCF-7 cells.** (A, B, C) MCF-7 cells were co-transfected with ATF6α cDNA, and control siRNA (SC) or siRNA targeting the specified genes. mRNA levels of DDIT3 were assessed by RT-PCR at 9 hours following the ectopic expression of ATF6α. Data presented are representatives of three independent experiments with similar results.

expression of ATF6α. Enriched functional annotation and canonical pathway analyses corroborated the involvement of cellular functions or pathways known for their crucial involvement in cellular senescence, supporting the validity of our analyses. Functional annotations such as ER stress response, asbestosis, stress response of cells and annotations related to cell death and apoptosis were identified from 6h-DEGs and 9h-DEGs. Canonical pathway analysis identified ER stress-related pathways and other pathways associated with cellular senescence, including the ID1 signaling pathway [21], PI3K signaling pathway [22], endocannabinoid cancer inhibition pathway [23, 24], NRF2-mediated oxidative stress response [25, 26], autophagy [27], and sirtuin signaling pathway [28, 29]. Notably, many senescence-related pathways were regulated within 9 hours following ATF6α ectopic expression.

Top scoring gene networks derived from 6h-DEGs and 9h-DEGs were associated with cell-to-cell signaling and interaction, featuring upregulated nodes such as *ATF6*, *ATF3* and *DDIT3*. These genes were also identified in enriched functional annotation and canonical pathway analyses. *DDIT3* emerged as a key player in cellular senescence induced by ectopic expression of ATF6α, as knockdown of *DDIT3* prevented cellular senescence. Also known as *GADD153* or *CHOP*, *DDIT3* belongs to the CCAAT/enhancer-binding protein (C/EBP) family transcription factor and is a well-established ER stress marker downstream of UPR activation pathways [30]. Despite its known role in inducing apoptotic pathways, *DDIT3* has been reported to promote senescence in alveolar epithelial cells [31] and in PCSK6 deficient H9c2 cells [32]. Moreover, it was demonstrated that *DDIT3* expression increases with age [33], but its direct relevance to human aging has not been conclusively established. Our findings suggest a potentially important role of *DDIT3*. *ATF3*, a stress-responsive transcription factor belonging to the ATF/cAMP response element-binding (CREB) protein family, has been implicated in reconstructing chromatin accessibility to facilitate cellular senescence [34]. However, conflicting reports on the role of *ATF3*, both promoting and inhibiting effect on cellular senescence, exist in the literature [34, 35], highlighting its context-dependent role in this process. In our model system, knockdown of *ATF3* did not exhibit any significant impact on cellular senescence. The top scoring gene network derived from 9h-DEGs also included well-known regulators of cellular senescence, such as PI3K [22], p38 MAPK [22] and IL-6 [36]. Additionally, the network identified from time-related-DEGs included TP53, a key gene associated with cellular senescence in responses to stress [37]. Taken together, among the six common denominator genes, *ATF6*, *DNAJB9*, *CHAC1*, *HERPUD1*, *DDIT3*, *and ATF3*, of 6-DEGs and 9-DEGs (Table 1), the ATF6, DDIT3, and ATF3 genes were implicated in cellular senescence. The roles of the

DNAJB9, CHAC1, HERPUD1 genes in senescence induction have not been reported but these genes are all involved in tumorigenesis in breast cancers [38–40].

Upstream regulator analysis with 9h-DEGs identified six regulators and eleven targets related to cytostasis and 'cell viability and cell death of connective tissue cells.' Among the regulators that were upregulated or predicted-to-be-activated, knockdown *MAP2K1/2*, *PDGFB* or *GPAT4* significantly prevented cellular senescence induced by ectopic expression of ATF6α. Knockdown of DDIT3, a target of MAP2K1/2 and GPAT4 also inhibited cellular senescence. MAP2K1/2, known as MEK1/2, functions as upstream kinases of ERK1/2, a pathway well-documented in cellular senescence [41, 42]. ERK1/2 has been reported to activate *DDIT3* expression [43, 44]. Our study also provides evidence that ERK1/2 is involved in cellular senescence induced by the ectopic expression of ATF6α, as shown by the prevention of cellular senescence with the chemical inhibitor Ravoxertinib (S3O and S3P Fig). GPAT4, or Lysophosphatidic Acid Acyltransferase Zeta, catalyzes the conversion of glycerol-3-phosphate to 1-acyl-sn-glycerol-3-phosphate (lysophosphatidic acid) by incorporating an acyl moiety at the sn-1 position of the glycerol backbone. Although the role of GPAT4 in cellular senescence remains largely unknown, a report suggests that silencing lysophosphatidic acid receptor 1 in mesenchymal stem cells confers resistance to cellular senescence [45]. Our study reveals that ATF6α overexpression or activation seems to activate MAP2K1/2 (and GPAT4), which in turn regulates the mRNA expression of DDIT3 or increases DDIT3 activity via MAP2K1/2-mediated ERK1/2 activation [44], underscoring the involvement of MAP2K1/2/GPAT4-DDIT3 pathway in cellular senescence.

Additionally, upstream regulator analysis also highlights a significant role of PDGF-BB as a regulator in ER stress/ATF6α-induced cellular senescence. PDGF-BB, a mitogenic growth factor composed of two B subunits of PDGF, has been shown to induce cellular senescence in certain cell types such as human dermal fibroblasts [46] and smooth muscle cells [47]. Knockdown of its predicted targets, PPP1R15A or IL-6, resulted in the attenuation of cellular senescence. However, inhibition of PDGF-BB did not seem to regulate the mRNA expression of PPP1R15A or IL-6, leaving the intermediate target responsible for PDGF-BB-induced senescence unknown. Further investigation is essential for a comprehensive understanding of the regulatory axis involving PDGF-BB.

GSEA analysis conducted with 6h-DEGs and 9h-DEGs unveiled enriched signaling pathways potentially driving cellular senescence upon ectopic expression of ATF6α. Specifically, GSEA with 6h-DEGs highlighted the importance of MYC and mTORC1 signaling in driving cellular senescence. A substantial body of literature supports the role of mTOR in cellular senescence [48, 49]. mTOR plays a pivotal role in promoting the senescence-associated secretory phenotype, and its inhibition has demonstrated to prevent cellular senescence induced by various cellular stresses. While oncogenic activation of MYC typically induces apoptosis as a failsafe mechanism against cellular transformation, it can also trigger cellular senescence. However, the literature presents both pro-senescent effects [50, 51] and anti-senescent effects [52] of MYC. GSEA analysis with 9h-DEGs indicated the enrichment of the interferon-alpha (IFN-α) response and the interferon-gamma (IFN-γ) response. Previous studies have revealed that IFN-γ [53] and IFN-α [54] can induce cellular senescence in various human cell types. IFN-γ not only induces cellular senescence but also leads to oxidative stress and DNA damage [55]. These results suggest that multiple signaling pathways might be directly or indirectly involved in inducing ER stress-mediated cellular senescence, and over time, changes in the main signaling pathways that induce cellular senescence may occur.

Comprehending the molecular mechanisms underlying ER stress-induced cellular senescence holds promise for developing new strategies in cancer therapy. Therapy-induced senescence (TIS) mediated through ER stress/UPR can play a pivotal role in promoting therapy

resistance, cancer progression and recurrence. Given the highly variable senescence phenotype depending on cell type, senescence-inducing stimuli, and temporal stages of cellular senescence [12], it becomes necessary to investigate TIS mechanisms individually for each cancer type and at different stages of cellular senescence. Our study contributes valuable insights into the transcriptomic profile and the mechanism of ER stress-induced cellular senescence in a breast cancer cell line, MCF-7, which has estrogen receptor (ER) (+)/PR(+)/HER2(−) phenotypes, specifically focusing on the early stage of cellular senescence. This research provides a foundational understanding that opens potential therapeutic avenues for breast cancer intervention.

## Supporting information

**S1 Fig. mRNA expression levels of selected genes in the 6h-DEGs and 9h-DEGs after ATF6α ectopic expression.** (A, B) MCF-7 cells were transfected with ATF6α cDNA or an empty vector DNA as a control. RT-PCR was conducted to assess the mRNA levels of the indicated genes selected from the 6h-DEGs (A) or 9h-DEGs (B) in MCF-7 cells after ATF6α ectopic expression. Data represent three independent experiments with similar results. Ctrl, control; ATF6 O/E, ATF6α overexpression.
(TIF)

**S2 Fig. Gene interaction networks of 9h-DEGs.** (A) Gene network of 9h-DEGs with a score of 24, associated with cell cycle, cellular development, and connective tissue development and function. (B) Gene network of 9h-DEGs with a score of 20, linked to cellular development, cellular growth and proliferation, and cellular movement. (C) Gene network of 9h-DEGs with a score of 18, related to hematological system development and function, humoral immune response, and lymphoid tissue structure and development. (D) Gene network of 9h-DEGs with a score of 8, associated with cell-to-cell signaling and interaction, hematological system development and function, and immune cell trafficking. The legend illustrates the relationship between molecules within the network and their activation state.
(TIF)

**S3 Fig. Validation of roles of regulators and targets identified through upstream regulator analysis in cellular senescence induced by ATF6α ectopic expression.** MCF-7 cells were co-transfected with ATF6α cDNA, and control siRNA (SC) or siRNA for the indicated genes. After 5 days, SA-β-gal staining was performed. (A, C, E, G) Light microscopic images represent three independent experiments. (B, D, F, H) Percentages of SA-β-gal (+) cells were plotted. (I, K, M) Light microscopic images represent three independent experiments. (J, L, N) Percentages of SA-β-gal (+) cells were plotted. Data represent the mean of triplicate determinations ± S.D. (O, P) MCF-7 cells were transfected with ATF6α cDNA or an empty vector and treated with 100 nM Ravoxertinib, an ERK inhibitor, or 500 nM Trametinib, a MEK1/2 inhibitor. After 5 days, SA-β-gal staining was performed. Data represent the mean of triplicate determinations ± S.D. Ctrl, control; ATF6 O/E, ATF6α overexpression. * $P<0.05$; ** $P<0.01$; *** $P<0.001$; ns, non-significant.
(ZIP)

**S4 Fig. Regulation of PPP1R15A and IL-6 mRNA expression by PDGF-BB at 9 hours after ATF6α ectopic expression in MCF-7 cells.** MCF-7 cells were co-transfected with ATF6α cDNA, and control siRNA (si-SC) or PDGFB siRNA. RT-PCR was conducted to assess mRNA levels of PPP1R15A and IL-6 at 9 hours after ATF6α ectopic expression. Data are representative of three independent experiments with similar results.
(TIF)

**S5 Fig. Overall Survival (OS) analysis of breast cancer patients with high and low expressions of ATF6α for the ATF6α-MAP2K1/2-DDIT3 axis.** The OS of breast cancer patients with high and low expressions of ATF6α was analyzed by Kaplan-Meier plotter (https://kmplot.com/analysis). (A) The OS curves of breast cancer patients with high and low expression levels of ATF6. (B) The OS curves of breast cancer patients with high and low expression levels of MAP2K1. (C) The OS curves of breast cancer patients with high and low expression levels of MAP2K2. (D) The OS curves of breast cancer patients with high and low expression levels of DDIT3. Hazard ratio (HR), log rank *P* values, and the number of patients (n) were indicated in the plots.
(TIF)

**S6 Fig. The original gel images of Fig 6A.** Data represnet the raw uncropped images of Fig 6A. M, Molecular weight markers; X, lanes not included in the figures.
(TIF)

**S7 Fig. The original gel images of Fig 6B.** Data represnet the raw uncropped images of Fig 6B. M, Molecular weight markers; X, lanes not included in the figures.
(TIF)

**S8 Fig. The original gel images of Fig 6C.** Data represnet the raw uncropped images of Fig 6C. M, Molecular weight markers; X, lanes not included in the figures.
(TIF)

**S9 Fig. The original gel images of the 6h lane of S1A Fig.** Data represnet the raw uncropped images of the 6h lane of S1A Fig. M, Molecular weight markers; X, lanes not included in the figures.
(TIF)

**S10 Fig. The original gel images of the 9h lane of S1A Fig.** Data represnet the raw uncropped images of the 9h lane of S1A Fig. M, Molecular weight markers; X, lanes not included in the figures.
(TIF)

**S11 Fig. The original gel images of the 6h lane of S1B Fig.** Data represnet the raw uncropped images of the 6h lane of S1B Fig. M, Molecular weight markers; X, lanes not included in the figures.
(TIF)

**S12 Fig. The original gel images of the 9h lane of S1B Fig.** Data represnet the raw uncropped images of the 9h lane of S1B Fig. M, Molecular weight markers; X, lanes not included in the figures.
(TIF)

**S13 Fig. The original gel images of S4 Fig.** Data represnet the raw uncropped images of S4 Fig. M, Molecular weight markers; X, lanes not included in the figures.
(TIF)

## Author Contributions

**Conceptualization:** Ju Won Kim, So-Hyun Bae, Mi-Ryung Han, Jeongwon Sohn.

**Data curation:** Ju Won Kim, So-Hyun Bae, Yesol Moon, Eun Kyung Kim, Yongjin Kim, Mi-Ryung Han, Jeongwon Sohn.

**Formal analysis:** Ju Won Kim, So-Hyun Bae, Yesol Moon, Eun Kyung Kim, Yongjin Kim, Mi-Ryung Han, Jeongwon Sohn.

**Funding acquisition:** Mi-Ryung Han, Jeongwon Sohn.

**Investigation:** Ju Won Kim, So-Hyun Bae, Yesol Moon, Eun Kyung Kim, Yongjin Kim.

**Methodology:** Ju Won Kim, So-Hyun Bae, Yesol Moon, Eun Kyung Kim, Yongjin Kim.

**Project administration:** Yun Gyu Park, Mi-Ryung Han, Jeongwon Sohn.

**Resources:** Ju Won Kim, So-Hyun Bae, Yesol Moon, Eun Kyung Kim, Yongjin Kim, Mi-Ryung Han, Jeongwon Sohn.

**Software:** Ju Won Kim, So-Hyun Bae, Yesol Moon, Eun Kyung Kim, Yongjin Kim, Mi-Ryung Han, Jeongwon Sohn.

**Supervision:** Yongjin Kim, Yun Gyu Park, Mi-Ryung Han, Jeongwon Sohn.

**Validation:** Ju Won Kim, So-Hyun Bae, Yesol Moon, Eun Kyung Kim, Yongjin Kim, Mi-Ryung Han, Jeongwon Sohn.

**Visualization:** Ju Won Kim, So-Hyun Bae, Yesol Moon, Eun Kyung Kim, Yongjin Kim, Mi-Ryung Han, Jeongwon Sohn.

**Writing – original draft:** Ju Won Kim, So-Hyun Bae, Mi-Ryung Han, Jeongwon Sohn.

**Writing – review & editing:** Eun Kyung Kim, Yun Gyu Park, Mi-Ryung Han, Jeongwon Sohn.

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
