## [Decision Letter · Decision Letter 0]

24 May 2024

PONE-D-24-15574Transcriptomic analysis of cellular senescence induced by ectopic expression of ATF6α in human breast cancer cellsPLOS ONE

Dear Dr. Sohn,

Thank you for submitting your manuscript to PLOS ONE. After careful consideration, we feel that it has merit but does not fully meet PLOS ONE’s publication criteria as it currently stands. Therefore, we invite you to submit a revised version of the manuscript that addresses the points raised during the review process.

We look forward to receiving your revised manuscript.

Kind regards,

Aniruddha Datta

Academic Editor

PLOS ONE

Journal Requirements:

"This work was supported by the Basic Science Research Program through the National Research Foundation of Korea (NRF) funded by the Ministry of Education (Grant 2018R1D1A1B07048901 to J.S.) and the Ministry of Science and Technology Information and Communication (Grant 2021R1F1A1063994 to J.S.) of the South Korean government, and the Korea University (Grant K1824361 to J.S.) and the Incheon National University Research Grant in 2021 (to M.-R. H.). "

Reviewers' comments:

Reviewer's Responses to Questions

**Comments to the Author**

1. Is the manuscript technically sound, and do the data support the conclusions?

Reviewer #1: Yes

Reviewer #2: Yes

Reviewer #3: Yes

Reviewer #4: Yes

2. Has the statistical analysis been performed appropriately and rigorously? 

Reviewer #1: N/A

Reviewer #2: I Don't Know

Reviewer #3: N/A

Reviewer #4: Yes

3. Have the authors made all data underlying the findings in their manuscript fully available?

Reviewer #1: Yes

Reviewer #2: Yes

Reviewer #3: Yes

Reviewer #4: Yes

4. Is the manuscript presented in an intelligible fashion and written in standard English?

Reviewer #1: Yes

Reviewer #2: Yes

Reviewer #3: Yes

Reviewer #4: Yes

5. Review Comments to the Author

Reviewer #1: This paper covers in great detail the research to study cellular senescence as triggered by various stressors. The paper also describes with ample clarity the entire sequence of steps performed to conduct the analysis. All the wet-lab methods employed for individual steps come with enough information to be able to replicate the study.

Overall, I am impressed by the authors’ approach and discussion to highlight the molecular mechnanisms underlying the ER stress0induced cellular senescence and the identification of key regulators like MAP2K 1/2 and GPAT4, and their downstream target DDIT3 in the induction of cellular senescence.

However, I would personally find it more reader-friendly and it would improve the chances of receiving more citations if the authors were to add more content specifically around the phenotypic manifestations of the 20 genes they found were significant in their time-series analysis and also find ways to weave a few lines on how their phenotypic manifestations might be inter-related. Such description can also be extended to offer the authors’ proposed strategies on how that might be used to provide more intuition around personalizing cancer therapies.

Overall, this is a good paper, and it would just add so much more value if the authors were to provide some additional content.

Also, I would much prefer the authors shared their NGS sequencing data files.

Reviewer #2: It requires little more details about the motivation of using 6h and 9h DEGs. It also lacks the details of how specifically MAP2K1/2-DDI3 axis influences cellular senescence via ATF6-alpha expression. The paper does not mention how other could have play some roles in linking the axis.

Reviewer #3: The authors address an important question on role of ATF6a in cellular senescence. However, the manuscript can be made robust by

1) Are the DEG the same if they work if a different cell line system?

2) The use ATF6a as a down stream O/E gene for their analysis. This is a little counterintuitive as they are over expressing the gene and hence this is to be expected.

3) Can they relate this to patient survival data between patients having high and low ATF6a expression levels? That will provide a nice translation link to their work.

4) The biological mechanism ATF6a ectopic expression is lacking and could be outside the scope of this work.

Reviewer #4: I think the article is well written, following are the comments to the authors:

1. Page 5: Line 62-64, please cite the study for the finding. If it is part of reference 1, please recite or reword the sentence as "our study also led us to discover that ATAF6_alpha..... senescence"

2. Line 68-70 : can the authors provide the rationale behind studying cell senescence in cancer cell lines and the broader impact of it ?

3. Lines 86-93 : introduces the importance of cell senescence in this study and should be discussed earlier . Either immediately after line 70 or the authors can briefly discuss why cell senescence is important to study in cancer cell in the paragraph of Line 59-70.

4. line 99-100 : Why did the authors pick MCF-7 cell line ? How did breast cancer subtype of ER+, PR+ , and glucocorticoid + factor into their analysis ?

5. Line 225 : Have any of these 6 genes been implicated in cell senescence before ?

6. PLOS authors have the option to publish the peer review history of their article (what does this mean?). If published, this will include your full peer review and any attached files.

Reviewer #1: **Yes: **Sangeeta Shukla, PhD

Reviewer #2: No

Reviewer #3: No

Reviewer #4: No

---

## [Author Response · Author response to Decision Letter 0]

8 Jul 2024

We have uploaded a cover letter named 'Response to Reviewers'

---

## [Decision Letter · Decision Letter 1]

19 Aug 2024

Transcriptomic analysis of cellular senescence induced by ectopic expression of ATF6α in human breast cancer cells

PONE-D-24-15574R1

Dear Dr. Sohn,

We’re pleased to inform you that your manuscript has been judged scientifically suitable for publication and will be formally accepted for publication once it meets all outstanding technical requirements.

Kind regards,

Aniruddha Datta

Academic Editor

PLOS ONE

Additional Editor Comments (optional):

Reviewers' comments:

Reviewer's Responses to Questions

**Comments to the Author**

1. If the authors have adequately addressed your comments raised in a previous round of review and you feel that this manuscript is now acceptable for publication, you may indicate that here to bypass the “Comments to the Author” section, enter your conflict of interest statement in the “Confidential to Editor” section, and submit your "Accept" recommendation.

Reviewer #1: (No Response)

Reviewer #4: All comments have been addressed

2. Is the manuscript technically sound, and do the data support the conclusions?

Reviewer #1: Yes

Reviewer #4: Yes

3. Has the statistical analysis been performed appropriately and rigorously? 

Reviewer #1: N/A

Reviewer #4: Yes

4. Have the authors made all data underlying the findings in their manuscript fully available?

Reviewer #1: Yes

Reviewer #4: Yes

5. Is the manuscript presented in an intelligible fashion and written in standard English?

Reviewer #1: Yes

Reviewer #4: Yes

6. Review Comments to the Author

Reviewer #1: I closely compared the previous and recent versions of your submissions, and still find answers to the feedback/comments earlier not satisfying. However, the work described in the paper is undoubtedly novel and interesting, and I am willing to approve.

Reviewer #4: The authors have addressed my comments sufficiently and I do not have any further changes to recommend. Good Luck with the submission.

7. PLOS authors have the option to publish the peer review history of their article (what does this mean?). If published, this will include your full peer review and any attached files.

Reviewer #1: No

Reviewer #4: No

---

## [Editor Report · Acceptance letter]

27 Aug 2024

PONE-D-24-15574R1 

PLOS ONE

Dear Dr. Sohn, 

I'm pleased to inform you that your manuscript has been deemed suitable for publication in PLOS ONE. Congratulations! Your manuscript is now being handed over to our production team.

Kind regards, 

on behalf of

Dr. Aniruddha Datta 

Academic Editor

PLOS ONE